# Evaluating COVID-19 Booster Vaccination Strategies in a Partially Vaccinated Population: A Modeling Study

**DOI:** 10.3390/vaccines10030479

**Published:** 2022-03-19

**Authors:** Clément R. Massonnaud, Jonathan Roux, Vittoria Colizza, Pascal Crépey

**Affiliations:** 1RSMS—U 1309, ARENES—UMR 6051, EHESP, CNRS, Inserm, Université de Rennes, 35043 Rennes, France; clement.massonnaud@inserm.fr (C.R.M.); jonathan.roux@ehesp.fr (J.R.); 2Biostatistics Unit, University Hospital Charles Nicolle, 76000 Rouen, France; 3Institut National de la Santé Et de la Recherche Médicale (INSERM), Institut Pierre Louis d’Épidémiologie et de Santé Publique (IPLESP), Sorbonne Université, 75014 Paris, France; vittoria.colizza@inserm.fr

**Keywords:** COVID-19, SARS-CoV-2, vaccination, vaccines, booster, modeling, epidemiology

## Abstract

Background: Several countries are implementing COVID-19 booster vaccination campaigns. The objective of this study was to model the impact of different primary and booster vaccination strategies. Methods: We used a compartmental model fitted to hospital admission data in France to analyze the impact of primary and booster vaccination strategies on morbidity and mortality, assuming waning of immunity and various levels of virus transmissibility during winter. Results: Strategies prioritizing primary vaccinations were systematically more effective than strategies prioritizing boosters. Regarding booster strategies targeting different age groups, their effectiveness varied with immunity and virus transmissibility levels. If the waning of immunity affects all adults, people aged 30 to 49 years should be boosted in priority, even for low transmissibility levels. Conclusions: Increasing the primary vaccination coverage should remain a priority. If a plateau has been reached, boosting the immunity of younger adults could be the most effective strategy, especially if SARS-CoV-2 transmissibility is high.

## 1. Introduction

Several highly effective COVID-19 vaccines have been made available worldwide since the end of 2020 [1,2,3,4,5,6,7]. In France, four vaccines have been authorized: *Comirnaty^®^* (Pfizer/BioNTech), *Spikevax^®^* (Moderna), *Vaxzevria^®^* (AstraZeneca), and *COVID-19 Vaccine Janssen* (Janssen-Cilag) [8] and used in vaccination campaigns to help mitigate the epidemic. The vaccination campaign started with the elderly population and at-risk population in January 2021, and was progressively extended to younger age groups down to 12 years of age. The vaccination coverage for people aged 65 years or more reached 60% in June 2021, and plateaued at the end of the summer at 85%. However, coverage remained low in specific senior age classes; for example, only 78% of 80+ received a full vaccination. The vaccination of individuals aged 18 to 64 started later on during the spring, and really took off in May for the 50–64 age groups, and June for the 18–49 age groups. The vaccination coverage increased rapidly, but then slowed down significantly by the end of the summer, at levels ranging from 70% (30–39 age group) to 85% (60–64 age group). The vaccination of the 12–17 age group started in July and reached 50% at the end of August [9].

Among the SARS-CoV-2 variants that spread worldwide, the variant of concern Alpha (VOC B.1.1.7) became dominant in metropolitan France in February 2021 [10], and was then replaced by the more transmissible Delta VOC (B.1.617.2), which has been dominant since July 2021 [10]. Several studies now suggest that the efficacy of vaccines against the Delta VOC is lower than against other variants [4,5,6,11]. Moreover, recent studies show that vaccine immunity, especially regarding infection and symptomatic form, is waning with time [11,12,13]. Therefore, many countries have recommended vaccine booster shots for all or part of their already vaccinated population. In France, only people aged 65 years or more (and specific people at risk) were initially eligible to receive a booster dose, administered at least 6 months after completion of the primary vaccination. Recommendations have recently extended this to the 18+ vaccinated population, similarly to other countries [14].

With the surging wave in Europe, and in the context of global vaccine supply constraints, it is important to assess the effectiveness of booster vaccination campaigns. The objective of this study was to analyze the impact, on hospital admissions and deaths, of COVID-19 primary and booster vaccination strategies targeting different age groups, under various hypotheses of vaccine efficacy and virus transmission levels, using France as a case study.

## 2. Methods

### 2.1. COVID-19 Transmission Model

We used a deterministic, age-structured, compartmental epidemic model, based on demographic and age profile of the population of metropolitan France. This is described in Appendix A and described in previous publications [15]. This model accounted for differences in susceptibility, severity and contacts across age groups [16,17,18]. To consider the impact of vaccines, we updated the model to a multi-branch transmission model, duplicating the main branch for each vaccine considered (Appendix A). In this framework, non-vaccinated individuals from susceptible and removed compartments could be vaccinated and integrated onto a vaccinated branch. This multi-branch model allowed us to use vaccine specific efficacies in each branch and, therefore, to model booster doses. The main outcomes returned by the model were the daily number of newly exposed and hospitalized individuals, among others.

### 2.2. Parametrization

The model is fitted dynamically on the number of observed hospitalizations from the start of the pandemic in March 2020 to September 2021 (Appendix A). We use a 7-day sliding window to estimate transmission parameters and shift the window by steps of 3 days. This method allows us to precisely reproduce epidemiological dynamics due to variations in transmissibility related to external factors such as mitigation measures or season. To account for the impact of VOCs on transmission parameters, we considered a 40% increase in the risk of hospitalization between Alpha VOC and wild type [19] and an 80% increase between Delta VOC and Alpha VOC [4]. Moreover, to take into account an increase in transmissibility of the virus due to seasonality during the simulated period, we progressively increased the transmission over two weeks, starting on 20 September 2021, and then held it constant until the end of the simulation (1 March 2022), to reproduce a 41% increase in winter time [20] of the average reproduction number estimated in August (R_eff_ = 1.0). In addition, we explored a range from 0% to 200% of increase in transmissibility due to seasonality. The model successfully reproduced the observed evolution of the epidemic in France, and estimates of the immune status of the population (Appendix A) are in accordance with what is reported by observational studies [21]. Details about the model and its parameterization are presented in Appendix A.

### 2.3. Vaccine Efficacies

Our transmission model allowed us to consider vaccine efficacies against three COVID-19 stages: infection, symptomatic cases and hospitalization. We considered that infected people who were vaccinated had the same risk of transmission as those non-vaccinated. We chose to group vaccines in two categories: mRNA vaccines (Comirnaty^®^ and Spikevax^®^) and vector vaccines (Vaxzevria^®^ and Janssen COVID-19 vaccine) due to the similarities in their characteristics and efficacies. We considered distinct efficacies according to the predominant VOC (Alpha or Delta) on the French territory and we only used efficacies after a complete vaccine scheme (2 doses for all vaccines except Janssen COVID-19 vaccine, which had one dose). Given the complexity of modeling the differential waning of efficacy against infection, symptomatic case and hospitalization, we used a simplified approach where each efficacy is reduced at a fixed time point (that is, on 15 July when we assumed the Delta VOC became dominant) and for specific age groups. Two scenarios were considered when the Delta VOC was predominant: (1) a decrease in vaccine efficacy only in 65 years and older, which would correspond to a case where this population was vaccinated first and is facing a waning of immunity, and (2) a decrease in all age groups, which would correspond either to a scenario where all age groups started the primary vaccination at the same time, and are facing a waning of immunity, or to a scenario where the immunity reduction is due to the Delta VOC itself, even for people recently vaccinated. Table 1 shows the values for the different vaccine efficacies conditioned to the predominant variant and the scenarios that were used in the simulations [11]. We assumed the booster dose confers an immunity against the Delta VOC equivalent to that observed against the Alpha VOC, which may be a conservative estimate compared to latest evaluations [22].

### 2.4. Vaccination Strategies

We assessed the impact of different vaccination strategies starting on 1 September 2021. Two distinct analyses are presented: (A) a comparison of vaccination strategies prioritizing primary vaccinations or booster shots in various proportions, and (B) a comparison of booster strategies targeting different age groups, in a context of plateauing primary vaccination. In both analyses, the effectiveness of each strategy is evaluated by the proportion of hospitalizations and deaths avoided, compared to an artificial reference scenario in which all vaccination (booster or primary) is stopped on 1 September 2021.

In the first analysis, the strategies differed in the prioritization of 200,000 daily available doses: they could be allocated either as first or second doses for primary vaccinations, or as booster doses for people already fully vaccinated. For both primary vaccinations and booster doses, we assumed a maximum vaccination coverage of 90% of the eligible population (note that this level had already been reached by 1 September for the primary vaccination of people aged 70 to 79). Once this plateau was reached for one of the targets (primary vaccinations or booster doses), the remaining doses were all allocated to the other one (Appendix A). The eligible population for primary vaccinations were individuals aged 12 years and older, and the people eligible for booster doses were individuals aged 65 years and older, in accordance with French recommendations [23]. We evaluated five strategies ranging from all available doses allocated to new vaccinations, to all available doses allocated to booster vaccination, by steps of 50 k doses from one allocation to another. To assess the impact on the results of the rate of vaccination, we performed sensitivity analyses considering a lower vaccination rate of 100,000 doses per day, and a higher vaccination rate of 400,000 doses per day.

In the second analysis, we simulated the administration of 10 million booster doses across five age groups: 30–49 years, 50–64 years, 30 years or older, 50 years or older and 65 years or older, starting 1 September 2021, at a rate of 200,000 doses per day. Primary vaccination was stopped on 1 September. We then analyzed the impact of different levels of increase in virus transmissibility due to seasonality from no increase to a 200% increase compared to the mean transmissibility level estimated during summer 2021 (R_eff_ = 1.0).

The model was run from March 2020 to the end of August 2021, calibrated to fit the number of daily hospitalizations observed in France over this period. Then, the different strategies were simulated from 1 September 2021 to 1 March 2022. We finally estimated the number of hospitalizations and deaths avoided in each vaccination or boosting strategy compared to the baseline scenario with no vaccination. All the analyses were performed using the *R* software (version 4.1.1). The code of the model, as well as all the code producing the results, is available upon request.

## 3. Results

### 3.1. Primary Vaccination Versus Booster Doses

Figure 1 shows the effectiveness of vaccination strategies varying in the allocation of 200,000 daily doses between primary vaccination and booster doses compared to the baseline scenario in which all vaccination is stopped on 1 September 2021. Strategies prioritizing primary vaccination were systematically more effective than strategies prioritizing boosters. If vaccine efficacy is decreased for all age groups, the increase in transmission during winter would lead to a wave of new hospitalizations, which would have been mitigated by any vaccination strategy (Figure 1B). The strategy allocating all doses in priority to primary vaccinations would have been the most effective, resulting in a 62% reduction in hospitalizations over the period, whereas the strategy allocating all doses in priority to booster shots would have resulted in a 47% reduction (Figure 1D); the other strategies having intermediate effectiveness. The analysis stratified by age group showed the same ranking of the strategies, even for 65 years and older, albeit with smaller differences (Figure 1F). Even when the protection conferred by vaccines is decreased only for people aged 65 years or older, strategies prioritizing the boosting of this population were the least effective to prevent hospitalizations (Figure 1A,C,E). The rankings of the strategies were identical regarding the number of deaths, and in the sensitivity analyses considering lower (100,000 doses per day) or higher (400,000 doses per day) vaccination rates (Appendix A).

### 3.2. Prioritization of Booster Doses

Figure 2 shows the impact on the hospital admissions due to COVID-19 of 10 million booster shots depending on the targeted age groups, over the period from 1 September 2021 to 1 March 2022, compared to a baseline scenario of no booster shots and no primary vaccination. If vaccine efficacy is decreased for all age groups (Figure 2B,D,F), boosting people aged 30 to 49 years would be the most effective strategy, reducing the total number of hospitalizations by 71%, while boosting people aged 65 years or older would lead to an 18% reduction only (Figure 2D). In the other scenario, in which the efficacy is decreased only for people aged 65 years or older, the strategy targeting the 30–49 age group was the most effective, with a 25% reduction. The strategy targeting those 30+, and that targeting those 65+, were next with similar levels of reduction in the total number of hospitalizations: 18% and 16%, respectively (Figure 2C). Figure 3 shows the same analyses regarding the deaths due to COVID-19. If vaccine efficacy is decreased for all age groups, boosting people aged 30 to 49 years would still be the most effective strategy (Figure 3D). However, if vaccine efficacy is decreased only for people aged 65 years or more, a boosting strategy targeting this population would be the most effective at reducing the total number of deaths (Figure 3C).

### 3.3. Impact of Variations of Transmissibility

Previous results were obtained considering a 41% increase in viral transmissibility during winter compared to the summer. As uncertainty still exists around the seasonality of SARS-CoV-2 transmission, we also evaluated whether the prioritization of vaccination strategies is sensitive to variations in transmissibility, either due to seasonality or loss of efficacy of contact reduction measures. When vaccine efficacy was decreased for all age groups, the ranking of the strategies did not change with changes in transmissibility, and targeting the 30–49 age group remained the most effective boosting strategy to reduce both hospital admissions and deaths (Figure 4B,D). However, when the vaccine efficacy was decreased only for people aged 65 years or older, the ranking of the strategies varied depending on the assumed winter viral transmissibility (Figure 4A,C). For a 30% transmissibility increase or more, relative to the mean transmissibility in summer (corresponding to R_eff_ = 1), targeting the 30–49 age group was the most effective boosting strategy to reduce hospitalizations. Regarding the proportion of deaths avoided, the differences between the strategies were smaller, and a larger degree of transmissibility (>80% increase) was needed to identify the 30–49 age group as the optimal group to target with boosters.

## 4. Discussion

Since a growing body of evidence shows that vaccine immunity to COVID-19 wanes with time [11,12,13] and decreases due to variants [4,5,6,11], several countries are implementing booster vaccination campaigns; however, data are lacking regarding the effectiveness of such strategies. In a first analysis, we compared vaccination strategies that differed by the prioritization of the doses between primary and booster vaccination. Strategies prioritizing primary vaccination were systematically more effective than strategies prioritizing boosters, in both scenarios of decreased vaccine efficacy. Overall, the most beneficial strategy was to allocate all available doses to primary vaccinations and then use the remaining doses as boosters. This is expected if vaccine doses are preferentially administered to senior age classes because of their higher risk of hospitalization. However, the strategy remains valid also when primary vaccination is extended to the younger age classes. This confirms that, although the baseline overall vaccination coverage was already relatively high, increasing the number of vaccinated people has a greater impact on the epidemic dynamics than boosting the protection of elderly individuals already fully vaccinated. The results stratified by age groups also revealed the same patterns. Even people aged 65 years or older would benefit more from a strategy prioritizing new vaccinations (which would necessarily target younger age groups) than from a strategy prioritizing their own boosting. These results reinforce the fact that population immunity plays a greater role in decreasing the risks of vulnerable populations than their individual protection, thanks to the overall reduction in viral circulation in the population. However, this result could be affected by the application of recommendations or restrictions that mostly impact active adults (e.g., telework, closure of restaurants and bars or curfews).

We also analyzed the impact of booster strategies targeting different age groups, in a case where primary vaccination has reached a plateau. In the scenario in which vaccine efficacy is decreased for all age groups, the most effective strategy in order to reduce both hospitalizations and deaths was to boost the immunity of the 30–49 age group. In the scenario in which the vaccine efficacy is decreased only for those 65 years and older, the different boosting strategies had varying effectiveness depending on the level of virus transmissibility in the population, and the outcome considered (hospitalizations or deaths). These results suggest that when deciding which populations to boost, two mechanisms must be considered: immunization of people with a high risk of severe forms of the disease, but also immunization of people with a high risk of transmission, which would reduce the viral circulation and therefore further protect the population at risk of severe consequences. Therefore, in the context of COVID-19, in which these two mechanisms involve different age groups, the optimal boosting strategy might be to target these two populations rather than only people above a certain age. Moreover, one must also consider the level of SARS-CoV-2 transmissibility in the population, which could be affected by seasonality, mitigation measures or new variants.

These results must be interpreted with caution as several assumptions and simplifications had to be made. Because of similarities between vaccines of the same type, we considered only two types of vaccines: mRNA vaccines and vector vaccines. This choice was made for simplification purposes, and because only one brand is dominant by vaccine type in France, with the Pfizer vaccine representing 88.6% of the vaccinations with mRNA vaccines, and the AstraZeneca vaccine 88.5% of the vaccinations with vector vaccines. Regarding the vaccine efficacies, we chose to consider distinct values depending on the predominant VOC on the French territory to account for the reduced efficacies against the Delta variant [5,6], and we considered two sets of vaccine efficacies against the Delta VOC, one in which the efficacies against Delta VOC are slightly lower than against Alpha VOC, and one in which the efficacies are greatly decreased for the protection against infection, according to recent studies [11,12,13]. We considered a first scenario in which the efficacies are decreased only for people aged 65 years and older, and a second scenario in which the efficacies are decreased for all age groups. However, we did not model the progressive decrease in vaccine efficacy over time but rather changed the parameters at fixed time points. Moreover, data on the protection granted by a booster dose are still scarce, so we assumed that the efficacies obtained after a booster dose were equal to the values with the Alpha variant. A recent study from Israel and the first results of clinical trials [24] seem to confirm this assumption.

The vaccination strategies were designed arbitrarily considering that 200,000 doses could be administered daily, until a plateau of 90% coverage was reached. This choice of 200,000 daily doses was based on the observed number of doses used in France for first and second injections at the end of August 2021. The choice of 90% for the vaccination coverage plateau was based on the observed vaccination coverage reached in the elderly population in France. Although these parameters, together with the levels of virus transmissibility during winter, directly affect the results, sensitivity analyses showed that they only change the impact of the vaccination strategies in terms of absolute numbers, but that the ranking of the strategies remained unchanged (Appendix A). Moreover, the parameters associated with the natural history of SARS-CoV-2 infection used here are those of the original virus strain. Preliminary results from Li et al. suggest that the average incubation time of the Delta variant could be as short as four days [25]. Finally, our model also does not allow for a recovered individual to be infected a second time by the SARS-CoV-2 virus. Although cases of re-infection have been documented [26], not much is known about the risk of re-infection after having recovered from COVID-19 [27]. At this stage, it is not likely to impact the results, but it could be added later on when more data become available.

The model presented here is able to simulate age-specific, complex vaccination strategies that combine primary vaccinations and booster shots in various proportions, and at different rates. The analyses were performed in the context of metropolitan France; however, we expect that the conclusions drawn here would hold in many similar settings; that is, countries where a portion of the population remains unvaccinated while the majority of the vaccinated individuals completed their vaccination four to five months before. While many countries have already announced plans for booster vaccine programs, others that struggle to achieve high coverage for the primary vaccinations have raised concerns regarding the equity in the access to vaccines. The WHO called into question the relevance of booster programs in the context of a pandemic where the majority of the world population has not received at least one dose [28]. The recent emergence of the Omicron VOC [29] illustrates how low vaccination coverage in large parts of the world might increase the risk that new, more potent variants emerging could, in turn, threaten even fully vaccinated countries. This work provides important insight in the matter of primary and booster vaccination, using the context of France as a case study, and can prove useful to inform these critical public health decisions.

Countries should continue pursuing primary vaccination in unvaccinated individuals to reduce hospitalizations and deaths. At the same time, with a rapid surge in cases in Europe due to the Delta variant and the threat of newly emerging variants, administering the booster doses to active adults would be overall more effective in mitigating the impact on the health system, under expected seasonal conditions of viral transmissibility.

## Figures and Tables

**Figure 1 vaccines-10-00479-f001:**
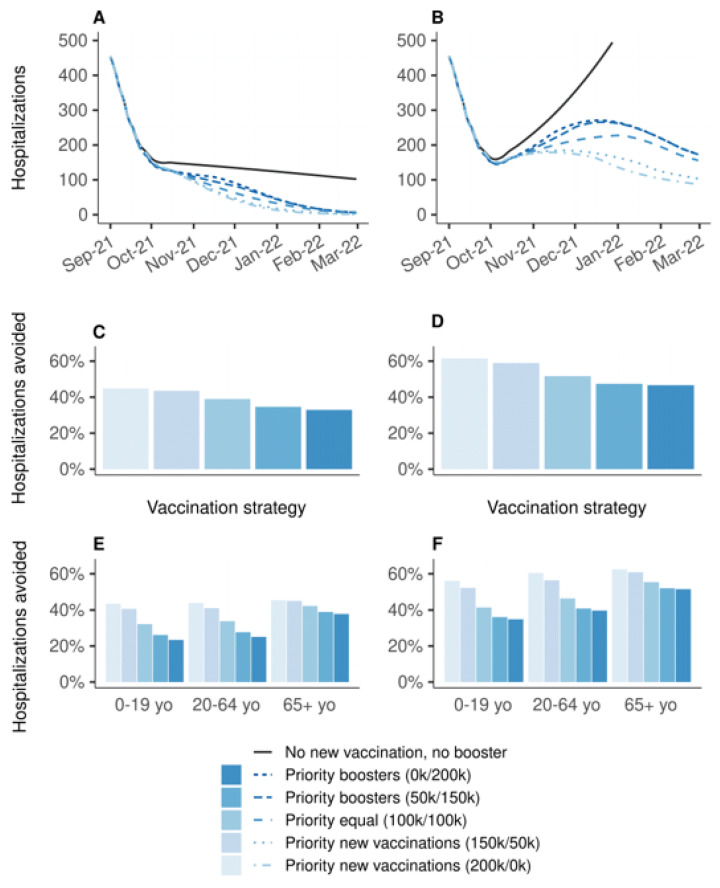
Effectiveness on hospitalizations of vaccination strategies varying in the allocation of 200,000 daily doses between primary vaccination and booster shots, over the period from 1 September 2021 to 1 March 2022, compared to a baseline scenario in which all vaccination is stopped on 1 September 2021. (**A**,**C**,**E**) Vaccine efficacy decreased only for people aged 65 years and older. (**B**,**D**,**F**) Vaccine efficacy decreased for all age groups. (**A**,**B**) Daily new hospitalizations, (**C**,**D**) proportion of hospitalizations avoided, (**E**,**F**) proportion of hospitalizations avoided by age group. A prioritization strategy of (150 k/50 k) means 150 k daily doses for primary vaccination and 50 k daily booster doses until 90% coverage of one target population is reached, then all 200 k daily doses are allocated to the other target population.

**Figure 2 vaccines-10-00479-f002:**
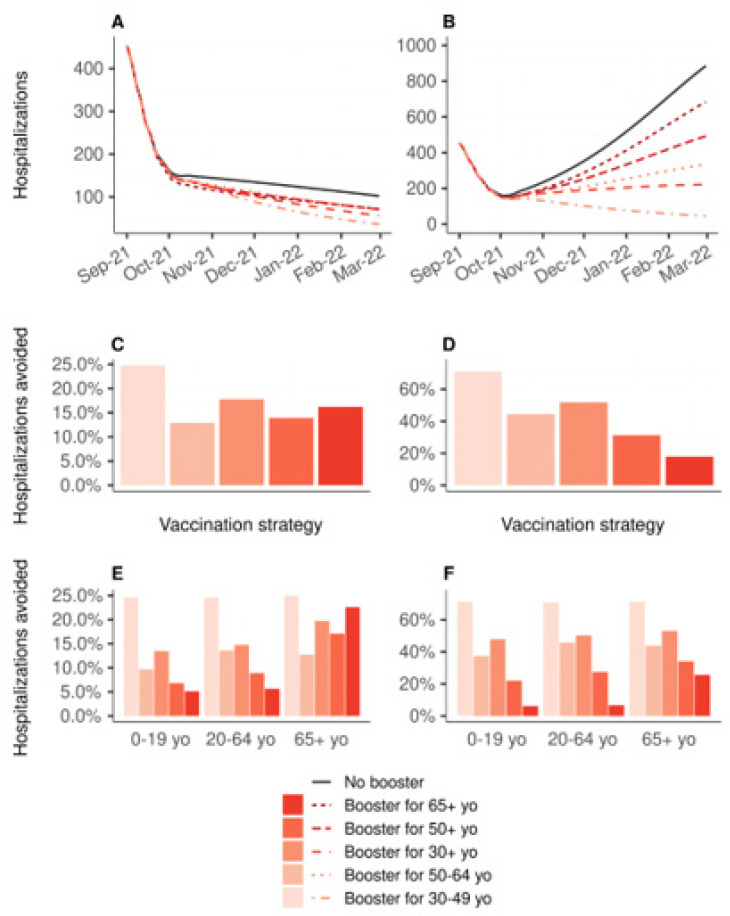
Impact on the hospital admissions of 10 million booster shots depending on the targeted age groups, over the period from 1 September 2021 to 1 March 2022, compared to a baseline scenario of no booster shots. In all strategies, primary vaccination is stopped on 1 September 2021. (**A**,**C**,**E**) Vaccine efficacy decreased only for people aged 65 years and older. (**B**,**D**,**F**) Vaccine efficacy decreased for all age groups. (**A**,**B**) Daily new hospitalizations, (**C**,**D**) proportion of hospitalizations avoided, (**E**,**F**) proportion of hospitalizations avoided by age group.

**Figure 3 vaccines-10-00479-f003:**
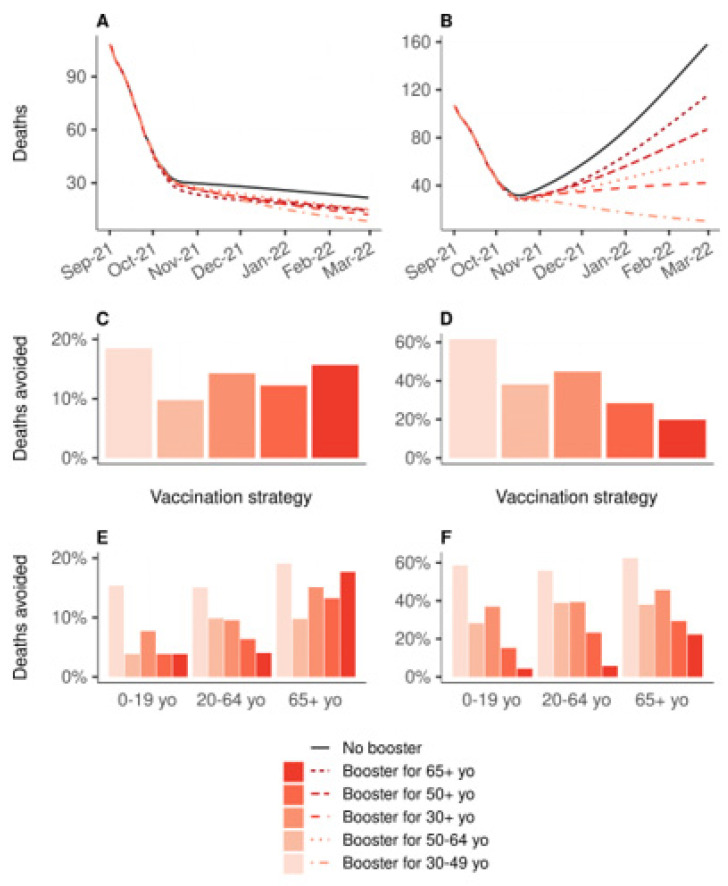
Impact on the number of deaths of 10 million booster shots depending on the targeted age groups, over the period from 1 September 2021 to 1 March 2022, compared to a baseline scenario of no booster shots. In all strategies, primary vaccination is stopped on 1 September 2021. (**A**,**C**,**E**) Vaccine efficacy decreased only for people aged 65 years and older. (**B**,**D**,**F**) Vaccine efficacy decreased for all age groups. (**A**,**B**) Daily deaths, (**C**,**D**) proportion of deaths avoided, (**E**,**F**) proportion of deaths avoided by age group. Note that only the booster strategy targeting the 30–49 yo had an impact on the percentage of deaths avoided in the 0 to 19 yo on inset (**E**) (one death avoided).

**Figure 4 vaccines-10-00479-f004:**
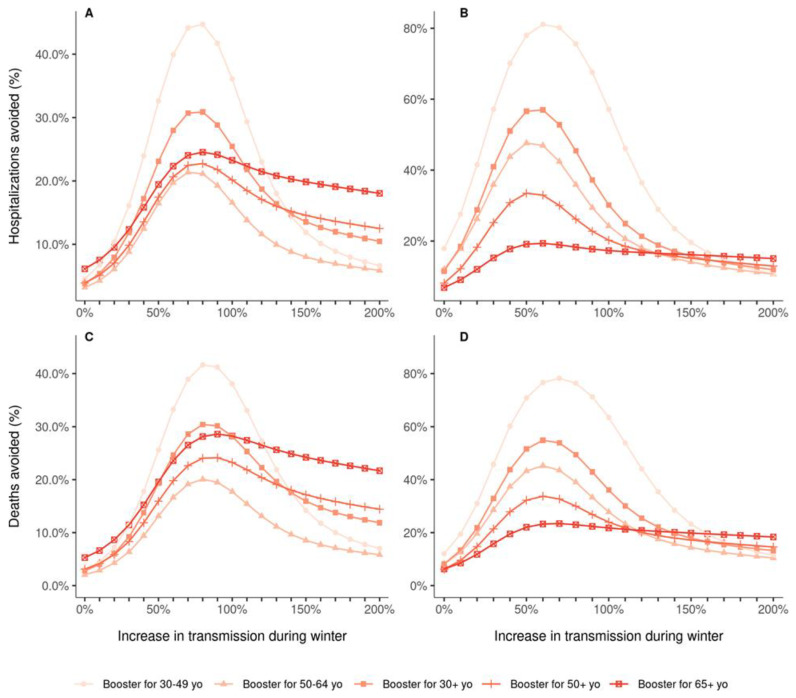
Proportion of hospitalizations (**A**,**B**) and deaths (**C**,**D**) avoided by 10 million booster shots compared to a baseline scenario of no booster shots, depending on the targeted age groups and the level of transmission during the winter, over the period from 1 September 2021 to 1 March 2022. In all strategies, primary vaccination is stopped on 1 September 2021. The increase in transmission starts on 20 September 2021 and is relative to the mean transmission levels during the summer 2021 (Reff = 1.0). (**A**,**C**) Vaccine efficacy decreased only for people aged 65 years and older. (**B**,**D**) Vaccine efficacy decreased for all age groups.

**Table 1 vaccines-10-00479-t001:** Retained efficacies against infection, symptomatic case and hospitalization for mRNA (Comirnaty^®^ and Spikevax^®^) and vector vaccines (Vaxzevria^®^ and Janssen COVID-19 vaccine) according to the predominant SARS-CoV-2 variant.

		Vaccine	Infection	Symptomatic Case	Hospitalization
Efficacy against Alpha VOC	Base efficacy	Vector vaccine	73% ^4^	75% ^5^	86% ^6^
		mRNA	92% ^4^	94% ^5^	97% ^7^
Efficacy against Delta VOC	Base efficacy	Vector vaccine	60% ^4^	67% ^5^	92% ^6^
		mRNA	79% ^4^	88% ^5^	96% ^6^
	Decreased efficacy	Vector vaccine	53% ^12^	53% ^12^	92% *
		mRNA	53% ^12^	53% ^12^	93% ^12^
	Efficacy after booster dose	mRNA	92%	94%	97%

* We assumed that the waning of vector vaccine efficacy was similar to that of mRNA vaccines, hence, no waning on efficacy against hospitalization. ^4,5,6,7,12^ The superscript numbers are actually the references from which the data was taken, found in the citations.

## Data Availability

The code of the model, as well as all the code producing the results, is available upon reasonable request.

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
