# Peer review of "Evaluating COVID-19 Booster Vaccination Strategies in a Partially Vaccinated Population: A Modeling Study"

_vaccines, 2022, doi:10.3390/vaccines10030479_

Round 1
Reviewer 1 Report
Interesting topic. Effectiveness of booster vaccine is an essential issure we face currently in all countries. Your mathematical method is valid but can be more readable if several points are corrected and specified as follows:
- In the method, you use compartment model, but you do not describe the differential/difference equations used for the model (not even in the supplementary material). Please specify these.
- You say on page 3 line 99 that "each efficacy is reduced at fixed time points and for specific age-groups". At which time does the decrease occur? or why not use exponential curve (i.e. λ * exp(-λt)) for better fit?
- In the result (section 3.1), you mention the "effectiveness" of vaccine strategies, while you mention the "efficacy" in the method section. These two terms are very different, but yet can be used interchangeably. Effectiveness refers to how much those that are vaccinated are protected from infection in the real world; i.e. herd immunity must be considered. On the other hand Efficacy is merely the difference in risk of getting infected in those vaccinated and unvaccinated givne the same exposure. Please make sure that these two terms are used correctly.
Author Response
The authors wish to sincerely thank the reviewer for the constructive comments. Here is our point-by-point response.
1. In the method, you use compartment model, but you do not describe the differential/difference equations used for the model (not even in the supplementary material). Please specify these.
Thank you very much for your comment, the equations were indeed missing although they are obviously an important aspect of the modelling work. They are now fully described in the supplementary materials.
2. You say on page 3 line 99 that "each efficacy is reduced at fixed time points and for specific age-groups". At which time does the decrease occur? or why not use exponential curve (i.e. λ * exp(-λt)) for better fit?
We agree with the reviewer that waning of efficacy is best modelled through an exponential curve. However, in this particular case the decrease in vaccine efficacy that we consider is only due to the emergence of a new variant with a better ability to evade the host immune response. In our analysis, we consider this reduction with the emergence of the Delta variant, which quickly became dominant in July 2021. Over such a short period, we do not think that using an exponential waning would be necessary as it is unlikely to change any of our result. We have specified in the manuscript the exact time on which the change is occuring.
3. In the result (section 3.1), you mention the "effectiveness" of vaccine strategies, while you mention the "efficacy" in the method section. These two terms are very different, but yet can be used interchangeably. Effectiveness refers to how much those that are vaccinated are protected from infection in the real world; i.e. herd immunity must be considered. On the other hand Efficacy is merely the difference in risk of getting infected in those vaccinated and unvaccinated givne the same exposure. Please make sure that these two terms are used correctly.
We agree with the reviewer that those two terms have different meanings, and should not be used interchangeably. We have thoroughly checked the manuscript for improper use of the terms efficacy/effectiveness. We have made sure to use "efficacy" when referring to the vaccine's characteristics, and "effectiveness" when referring to the population impact of the vaccination strategies. We have made the corrections in the manuscript accordingly.
Reviewer 2 Report
The paper is very interesting but I ask to the authors to explain better the use of Astra Zeneca and Janssen COVID 19 vaccines and a possible different responses between vaccination schedule and booster in immunocompromised patients. I think that the paper is fit for the publication in the journal after a complete response to required questions.
Author Response
The authors wish to sincerely thank the reviewer to have accepted to review this work. Here is our response to the reviewer's question.
The paper is very interesting but I ask to the authors to explain better the use of Astra Zeneca and Janssen COVID 19 vaccines and a possible different responses between vaccination schedule and booster in immunocompromised patients. I think that the paper is fit for the publication in the journal after a complete response to required questions.
In our analysis, we only differentiate mARN based vaccines from Adenovirus based ones. Hence, we do not account for possible differences between AstraZeneca and Janssen COVID-19 vaccines. In addition, in France as of March 2022, AstraZeneca and Janssen represent 7.8M and 1M doses out of more than 140M injected doses, and none of them has been used as a booster. Hence, we do not expect any specific impact of those vaccines on our results.
We understand the importance and specificity of vaccination schedules among immunocompromised patients. Unfortunately our model assesses public health outcomes at the population level for each age-group defined. We could not directly account for specific groups of high risk patients, nor do we think that we would have been able to generate relevant results for this group with this “general population” approach.
Reviewer 3 Report
In the current manuscript, authors perform an analysis regarding the population impact of primary COVID 19 vaccination by comparison with booster administration in different age groups.
The topic is of high interest since the COVID 19 pandemic is ongoing and different countries apply different strategies of vaccination. This comparison could aid the scientific community to apply the most effective measures for population protection, and clinical data based on population from different EU countries are important.
Conclusions and references provided are in agreement with the model used for data analysis.
Figures are explanatory for results
The manuscript is very interesting and the analysis is sound. A spelling check is recommended (eg: waning - waining?)
Author Response
The authors wish to sincerely thank the reviewer for the kind comments. We have performed a thorough spell check as recommended, and corrected all mistakes to the best of our abilities.